# The Interaction between Intratumoral Microbiome and Immunity Is Related to the Prognosis of Ovarian Cancer

Dashuang Sheng,[a,b] Kaile Yue,[a,b] Hongfeng Li,[a,b] Lanlan Zhao,[a,b] Guoping Zhao,[a,b,c,d] Chuandi Jin,[a,b] Lei Zhang[a,b,c]

aDepartment of Biostatistics, School of Public Health, Cheeloo College of Medicine, Shandong University, Jinan, China

bMicrobiome-X, National Institute of Health Data Science of China & Institute for Medical Dataology, Cheeloo College of Medicine, Shandong University, Jinan, China

cState Key Laboratory of Microbial Technology, Shandong University, Qingdao, China

dCAS Key Laboratory of Computational Biology, Bio-Med Big Data Center, Shanghai Institute of Nutrition and Health, University of Chinese Academy of Sciences, Chinese Academy of Sciences, Shanghai, China

Dashuang Sheng, Kaile Yue, and Hongfeng Li contributed equally to this work. Author order was determined based on their contribution.

**ABSTRACT** Microbiota can influence the occurrence, development, and therapeutic response of a wide variety of cancer types by modulating immune responses to tumors. Recent studies have demonstrated the existence of intratumor bacteria inside ovarian cancer (OV). However, whether intratumor microbes are associated with tumor microenvironment (TME) and prognosis of OV still remains unknown. The RNA-sequencing data and clinical and survival data of 373 patients with OV in The Cancer Genome Atlas (TCGA) were collected and downloaded. According to the knowledge-based functional gene expression signatures (Fges), OV was classified into two subtypes, termed immune-enriched and immune-deficient subtypes. The immune-enriched subtype, which had higher immune infiltration enriched with CD8$^+$ T cells and the M1 type of macrophages (M1) and higher tumor mutational burden, exhibited a better prognosis. Based on the Kraken2 pipeline, the microbiome profiles were explored and found to be significantly different between the two subtypes. A prediction model consisting of 32 microbial signatures was constructed using the Cox proportional-hazard model and showed great prognostic value for OV patients. The prognostic microbial signatures were strongly associated with the hosts' immune factors. Especially, M1 was strongly associated with five species (*Achromobacter deleyi* and *Microcella alkaliphila*, *Devosia* sp. strain LEGU1, *Ancylobacter pratisalsi*, and *Acinetobacter seifertii*). Cell experiments demonstrated that *Acinetobacter seifertii* can inhibit macrophage migration. Our study demonstrated that OV could be classified into immune-enriched and immune-deficient subtypes and that the intratumoral microbiota profiles were different between the two subtypes. Furthermore, the intratumoral microbiome was closely associated with the tumor immune microenvironment and OV prognosis.

**IMPORTANCE** Recent studies have demonstrated the existence of intratumoral microorganisms. However, the role of intratumoral microbes in the development of ovarian cancer and their interaction with the tumor microenvironment are largely unknown. Our study demonstrated that OV could be classified into immune-enriched and -deficient subtypes and that the immune enrichment subtype had a better prognosis. Microbiome analysis showed that intratumor microbiota profiles were different between the two subtypes. Furthermore, the intratumor microbiome was an independent predictor of OV prognosis that could interact with immune gene expression. Especially, M1 was closely associated with intratumoral microbes, and *Acinetobacter seifertii* could inhibit macrophage migration. Together, the findings of our study highlight the important roles of intratumoral microbes in the TME and prognosis of OV, paving the way for further investigation into its underlying mechanisms.

**KEYWORDS** gynecological, ovarian cancer, tumor microenvironment, microbiome, prognostic biomarkers

Address correspondence to Lei Zhang, zhanglei7@sdu.edu.cn, or Chuandi Jin, jinchuandi@sdu.edu.cn.

The authors declare no conflict of interest.

Ovarian cancer (OV) is one of the three most common gynecological malignancies, with the highest mortality. In 2020, there were 313,959 new cases and 207,252 deaths from ovarian cancer worldwide (1). Despite improvements in survival rates over the last 40 years, three fifths of women still die within 5 years of diagnosis (2). There is an urgent need to identify factors associated with the prognosis and therapeutic effect of OV patients.

The tumor microenvironment (TME) plays a significant role in clinical outcomes and response to therapy (3–6). Immune cells such as macrophages, lymphocytes, and natural killer cells are the important components of the TME, which can profoundly influence tumor progression and the efficacy of anticancer therapies. The intratumoral M1 macrophages and CD8$^+$ T cells have been demonstrated to play a vital role in the prognosis of ovarian cancer treatment (5, 7). Previous studies have classified the TME according to immune status (3, 4) and found that the immune-enriched subtypes were associated with a better prognosis than subtypes with a lack of immune infiltration. Although the influence of immune infiltration on cancer therapy and prognosis has been broadly investigated (3, 5–7), the factor influencing immune infiltration is largely unknown.

The TME is an attractive niche for microbial growth, and microbes have been identified within tumors for over a century (8). Most human cancer types harbor molecular evidence of an intratumoral microbiota (8, 9), including bacterial communities located peripheral to and deep within tumor tissues. The microbiota influence oncogenesis and cancer treatment in many ways, not merely by acting as direct carcinogens. Recent studies showed that microbiota may impact the TME, including inflammatory mediators such as tissue-resident and peripherally recruited immune cells, fibroblasts, endothelial cells, adipocytes, and pericytes (10, 11). It has been reported that the cervicovaginal microbiome is implicated in ovarian cancer risk (12). The existence of intratumoral microbiota in OV has been proven recently (8, 13, 14), and it has been reported that the distribution of bacteria differs between cancerous and noncancerous ovarian tissues *in situ*. However, whether intratumoral microbes are associated with the TME and prognosis in OV is still unknown. Besides, which functional pathways of intratumoral microbes are associated with OV needs exploration. The whole-transcriptome sequencing (RNA-Seq) data provided by The Cancer Genome Atlas (TCGA) offered a good opportunity to explore the interaction between intratumoral microbiota and the host's gene expression (9, 15–18). Here, we extracted nonhuman reads from RNA-Seq data of TCGA-OV for microbiome analysis, assessed the association between intratumoral microbes and OV prognosis, and explored the relationships between tumor immunity and intratumoral microbes' taxa and/or functions. We found that the microbiota profile was different between immune-enriched and -deficient TMEs. Intratumoral microbiota was an independent predictor for survival prognostic of OV. Integrated analysis and cell experiments showed that tumor immunity was strongly associated with microbes.

## RESULT

**Two immune subtypes of ovarian cancer based on Fges have different TME characteristics.** Clustering analysis revealed that the molecular profiles of OV can be clustered into two distinct TMEs, termed the immune-deficient subtype (clust1) and the immune-enriched subtype (clust2), according to 29 knowledge-based functional gene expression signatures (Fges) (4) (Fig. 1A; see Fig. S1 in the supplemental material). Differential analysis revealed that baseline clinical indicators had no significant difference between clust1 and clust2 (Table 1). Patients in clust2 had higher stromal and immune scores with less tumor purity ($P < 0.05$) (Fig. 1B), higher proportions of CD8$^+$ T cells, activated CD4$^+$ memory T cells, M1 type of macrophages (M1), regulatory T cells, resting dendritic cells, and resting mast cells with fewer M0 type of macrophages ($P < 0.05$) (Fig. 1C), a higher tumor mutational burden (TMB) and a higher number of neoantigens, i.e., somatic nonsynonymous coding single nucleotide variants, immunogenic mutations, and potential neoantigenic peptides ($P < 0.05$) (Fig. 1D), and higher T cell receptor (TCR) and B cell receptor (BCR) diversity ($P < 0.01$) (Fig. 1E) than patients in clust1. Differentially expressed genes (DEGs) were significantly enriched in pathways related to immune activation, regulation, and diseases caused by immune abnormalities based on Gene Ontology (GO) and Kyoto Encyclopedia of Genes (KEGG) enrichment analysis (Fig. 1F and G). Moreover,

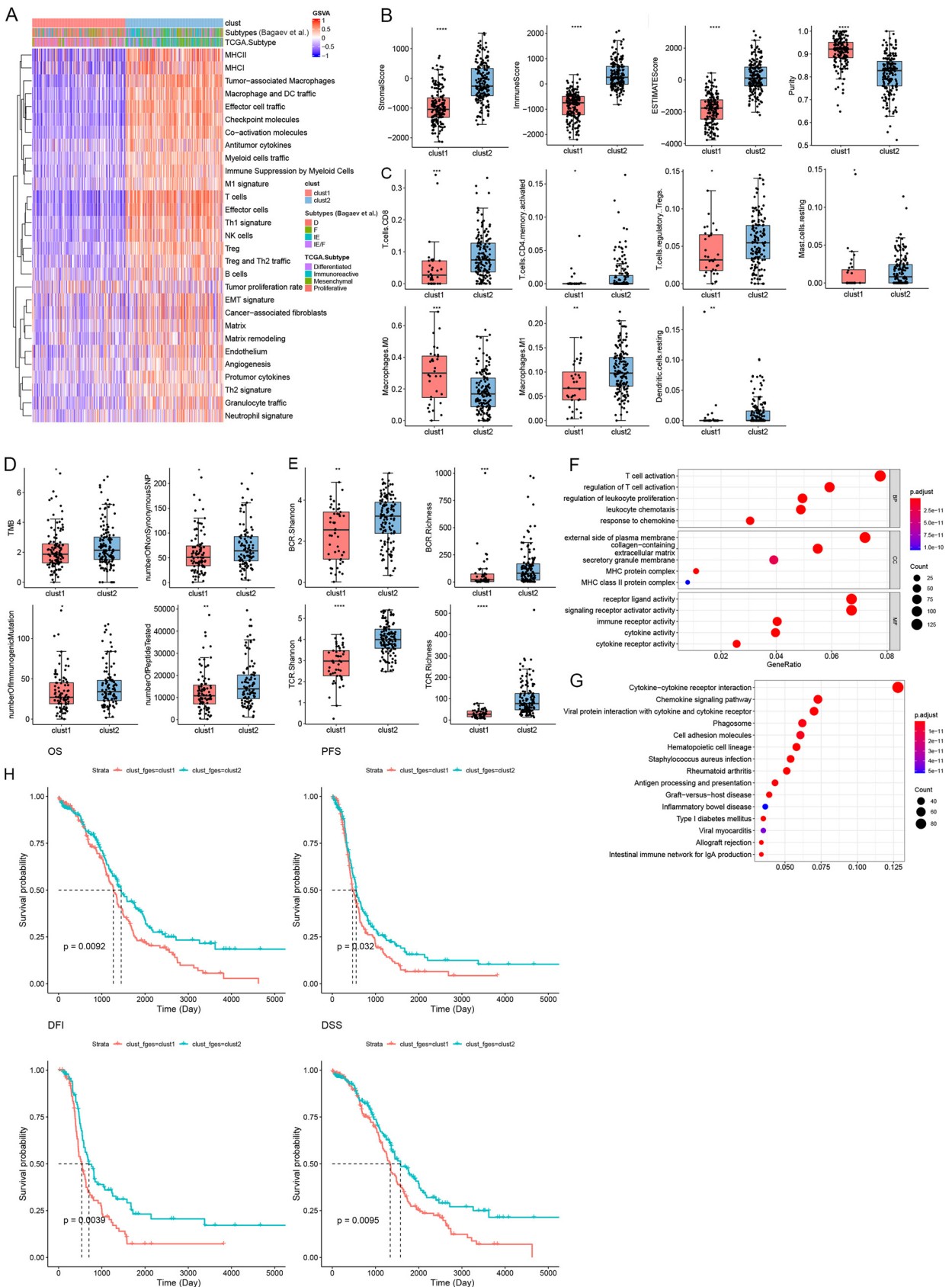

**FIG 1** Two ovarian cancer microenvironment subtypes were revealed by unsupervised analysis of the Fges. (A) Heat map of 373 TCGA ovarian serous cystadenocarcinomas (TCGA-OV) classified into two distinct TME subtypes based on k-means clustering of the 29 Fges. Additional

**TABLE 1** Patient baseline characteristics

| Characteristic | Value[a] for: | | P value |
| --- | --- | --- | --- |
| | clust1 ($n = 182$) | clust2 ($n = 191$) | |
| Age at index (yr) | 60.2 (11.3) | 59.0 (11.5) | 0.346 |
| Clinical stage | | | 1.000 |
| I/II | 10 (5.65) | 11 (5.91) | |
| III/IV | 167 (94.4) | 175 (94.1) | |
| Ethnicity | | | 0.478 |
| Hispanic or Latino | 5 (4.90) | 3 (2.59) | |
| Not Hispanic or Latino | 97 (95.1) | 113 (97.4) | |
| Pharmaceutical treatment | | | 0.504 |
| No | 3 (1.71) | 6 (3.23) | |
| Yes | 172 (98.3) | 180 (96.8) | |
| Radiation treatment | | | 0.318 |
| No | 163 (92.1) | 178 (95.2) | |
| Yes | 14 (7.91) | 9 (4.81) | |

[a]Values for categorical variables are numbers (%) of patients, while age at index is given as the mean (SD).

survival analysis showed that the patients in clust2 had a better prognosis for overall survival (OS, $P = 0.009$), progression-free survival (PFS, $P = 0.032$), disease-specific survival (DSS, $P = 0.009$), and disease-free interval (DFI, $P = 0.004$) (Fig. 1H).

In summary, based on their TME characteristics, OV patients can be clustered into immune-deficient and -enriched subtypes termed clust1 and clust2, respectively. clust1 is characterized by the lack of immune infiltration and high tumor purity, while clust2, on the other hand, exhibits high immune infiltration and high TMB, which is associated with better prognosis.

**Different TME subtypes exhibited different intratumoral microbiota profiles.** Of $103 \times 10^9$ sequencing reads in TCGA-OV, 6.4% were classified as nonhuman reads, and a total of 30 phyla, including 6 archaea and 24 bacteria, were identified, with *Proteobacteria* being the most abundant phylum and *Pseudomonas* being the most abundant genus (Fig. 2A and B). Although there was no significant difference in alpha-diversity ($P > 0.05$) (Fig. S2A), beta-diversity analysis showed that intratumoral microbial profiles were different between clust1 and clust2 ($P = 0.02$) (Fig. 3A) and beta-diversity was more dissimilar among individuals in clust1 ($P = 0.003$) (Fig. 3B). Based on linear discriminant analysis effect size (LEfSe) analysis, 58 species were enriched for clust1 that were mainly from *Pseudomonas*, while 11 species were enriched for clust2 (Fig. 3C). The differential species were closely associated with Fges (Fig. 3D). Under a different decontaminated standard, these results were consistent (Fig. S2B to D). Moreover, the microbial cooccurrence network of cluster1 was dominated by *Pseudomonas*, with *Pseudomonas brassicacearum* and *Pseudomonas* sp. Ost2 being the most crucial species. Only significant correlations ($P < 0.01$, $R^2 > 0.75$) based on the bootstrapping of 50 iterations were plotted (Fig. S3).

Functional analysis showed that microorganisms in clust1 and clust2 were enriched for different KEGG pathways. More pathways were enriched in clust1, i.e., taurine and hypotaurine metabolism (map00430), PI3K-Akt signaling pathway (map04151), and phosphatidylinositol signaling system (map04070) (Fig. 3E).

**FIG 1** Legend (Continued)
annotation includes the subtypes (immune enriched and fibrotic [IE], immune enriched and nonfibrotic [IE/F], fibrotic [F], and immune depleted [D]) taken from Bagaev et al. and TCGA subtypes "immunoreactive," "mesenchymal," "proliferative," and "differentiated" taken from the Cancer Genome Atlas Research Network. (B to E) Box plots showing differences in tumor cellularity (B), immune cells (C), neoantigens and TMB (D), and TCR and BCR richness and Shannon index (E). (F and G) Top 15 pathways determined by GO (F) and KEGG (G) enrichment analysis of differential expression genes between clust1 and clust2. (H) Kaplan-Meier curves of OS, PFS, DSS, and DFI in OV patients stratified by TME subtype classification. *, $P < 0.05$; **, $P < 0.01$; ***, $P < 0.001$.

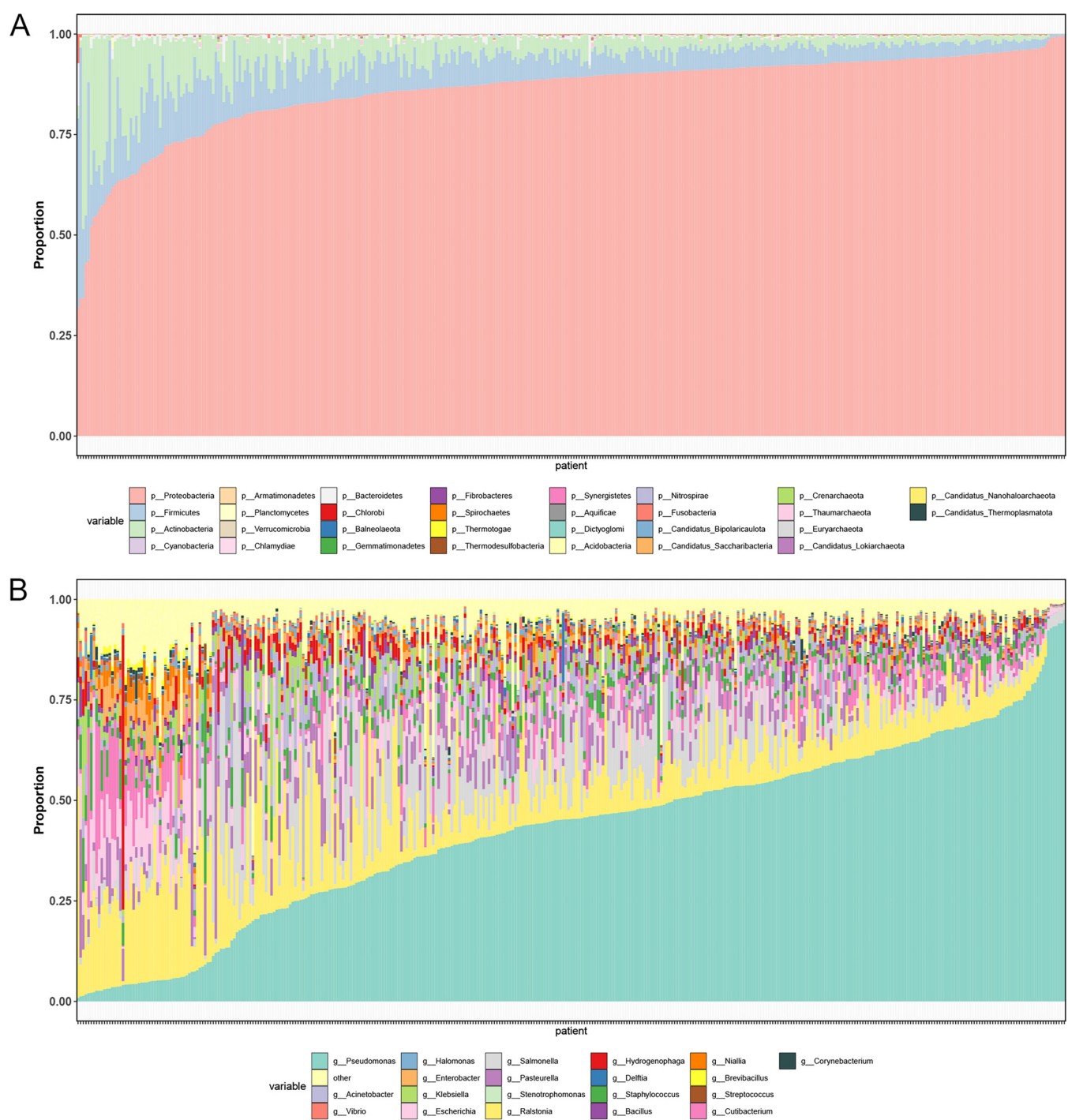

**FIG 2** Relative abundance data for each sample in the raw microbiome composition data. (A) Phylum; (B) genus.

Taken together, the intratumoral microbial profiles were significantly different between clust1 and clust2, which may be associated with the TME of patients with OV.

**Intratumoral microbiome was associated with OV prognosis.** Based on univariate Cox proportional-hazard model analysis, we identified 736 species as potential prognosis biomarkers of OV (Fig. 4A). Among these, (i) the abundances of 193 species were significantly associated with OS, of which 183 were risk factors (hazard ratio [HR] > 1; $P < 0.05$), while 10 were protective factors (HR < 1; $P < 0.05$); (ii) 219 species were significantly associated with DSS, of which 205 were risk factors and 14 were protective factors; (iii) 318 species were significantly associated with PFS, of which 194 were risk factors and 125 were

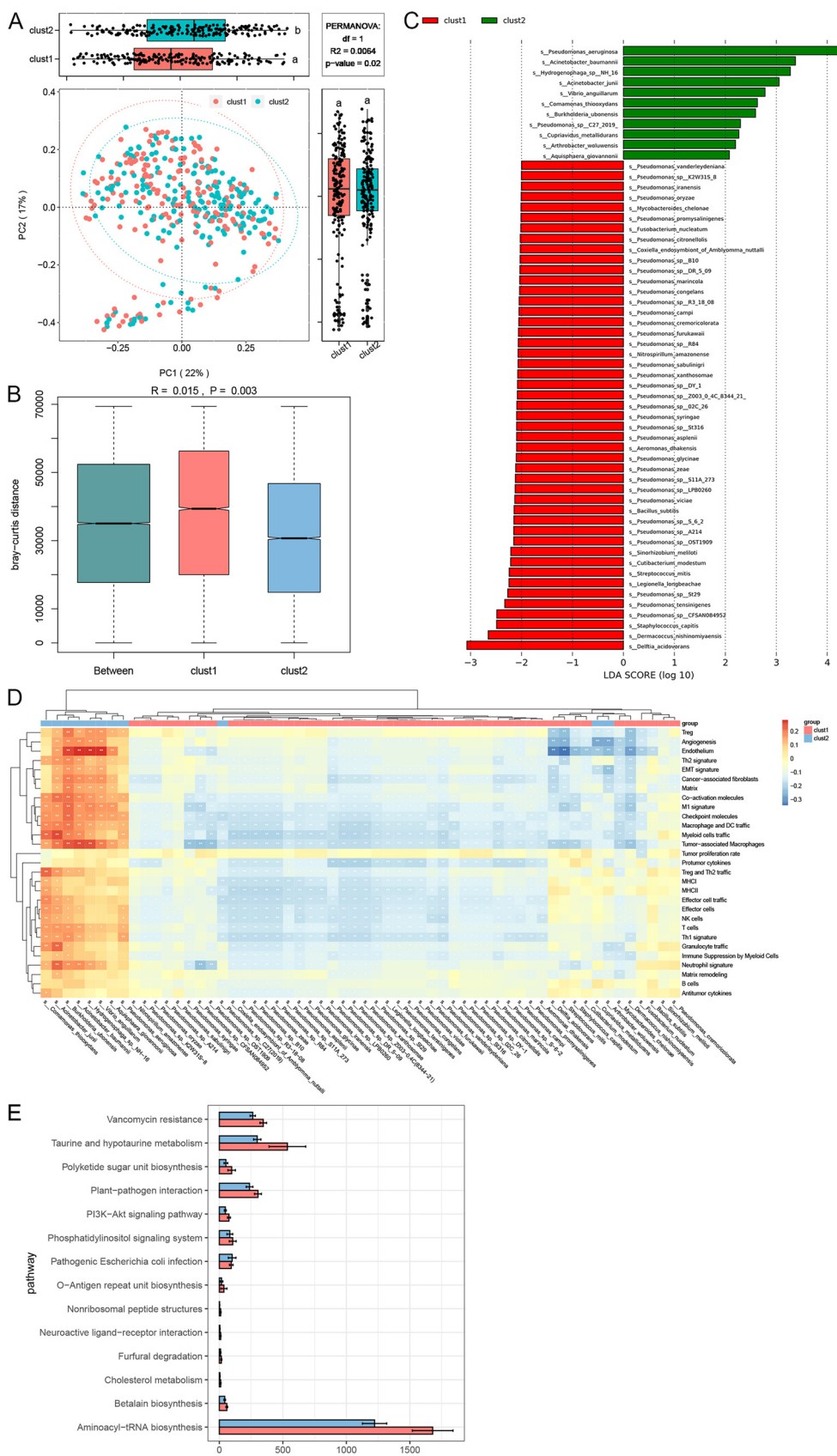

**FIG 3** clust1 and clust2 exhibited different intratumoral microbiota profiles. (A) PCoA and box plot shown along the first two principal coordinates on the Bray-Curtis distances for likely decontamination microbiome composition data.

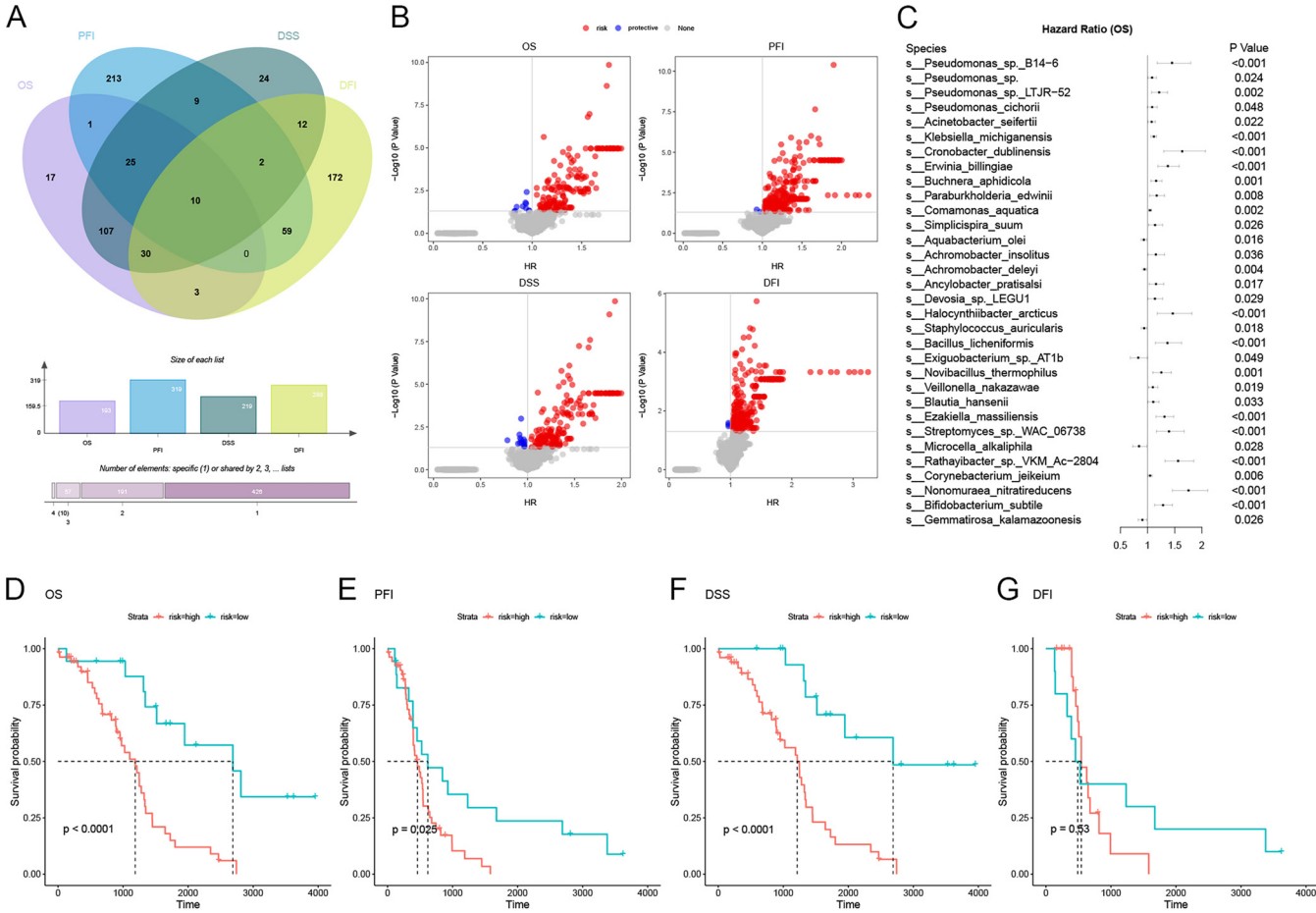

**FIG 4** The intratumoral microbiome was associated with OV prognosis. (A) Venn plot and histogram of prognostic species for OS, PFS, DSS, and DFI. (B) Volcano plot depicting the association between microbial abundance and OS, DSS, PFS, and DFI (red, risk species; blue, protective species). (C) Forest plot for hazard ratio from Cox proportional-hazard model constructed by LASSO-penalized Cox regression method (segments represent 95% confidence interval). (D to G) Kaplan-Meier curves of OS (D), DSS (E), PFS (F), and DFI (G) according to the level of risk score calculated for 32 prognostic species; the *P* value was calculated by the log rank test.

protective factors; and (iv) 288 species were significantly associated with DFI, of which 285 were risk factors and 3 were protective factors (Fig. 4B; Table S1 to S4). Based on microbial biomarkers associated with OS, a prediction model consisting of 6 protective species, 26 risk species, and age at diagnosis was constructed using the least absolute shrinkage and selection operator (LASSO)-penalized Cox regression method, which could well predict the overall survival probability of OV patients in the test set (C-index = 0.727) (Fig. 4C). After the test set was divided into high- and low-risk groups according to the lower quartile of risk score obtained by the Cox model, the survival time in the low-risk group was longer than that in the high-risk group (Fig. 4D). Prognostic models consisting of microbes also showed great prognostic value for PFS and DSS in OV patients (Fig. 4E to G). The presence or absence of these prognostic microbes was also significantly associated with OS, DSS, PFS,

**FIG 3 Legend (Continued)**
Ellipses represent the 95% confidence interval around the group centroid. *P* and $R^2$ values were determined by PERMANOVA. $R^2$ is the coefficient of determination, which represents the percentage of the total variance of the distance matrix explained by the tumor immune condition. The *P* value is the possibility that the distribution will be the same between the groups. (B) Bray-Curtis distances of the microbial communities among individuals within the same group and between different groups. *P* and *R* values were determined by ANOSIM. *R* is the difference in the distance ranks between groups and within groups. (C) Significant differentially abundant taxonomic biomarkers between clust1 and clust2 identified by LEfSe. (D) Correlation heat map between taxonomic biomarkers and 29 Fges. (*, *P* < 0.05; **, *P* < 0.01). (E) Histogram of KEGG pathways that are significantly different between clust1 and clust2. Mean and standard error of the mean (SEM) are shown.

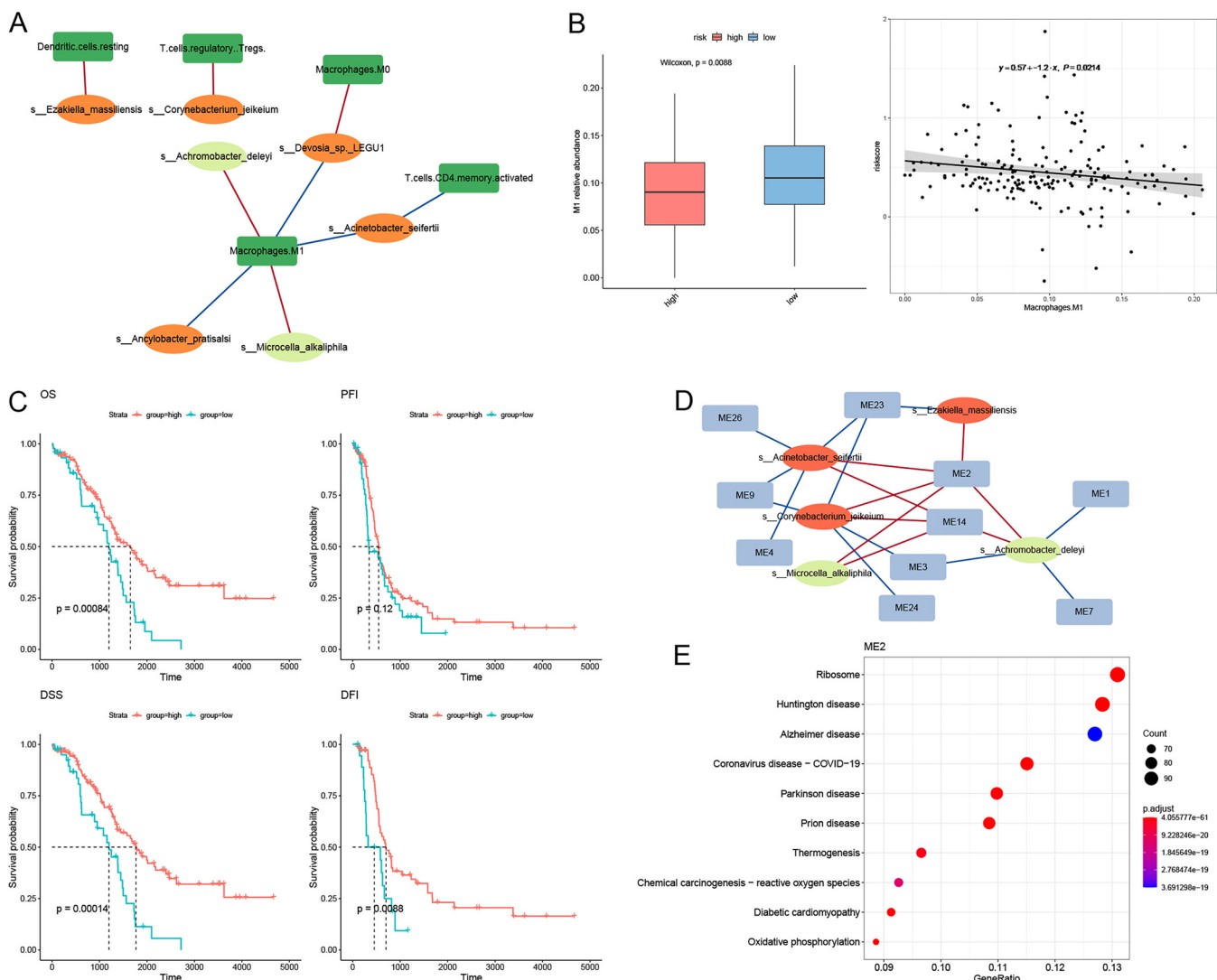

**FIG 5** Joint analysis of multiomics reveals potential host-immune interactions. (A) Correlation network for microbes and immune cells obtained from CIBERSORT (red, positive correlation; blue, negative correlation). (B) Box plot showing difference in M1 between high- and low-risk groups (left) and scatterplot of risk score showing negative association with proportion of M1 (right). (C) Kaplan-Meier curves of OS, DSS, PFS, and DFI according to the proportion of M1. (D) Correlation network for microbes and immune gene modules (MEs) obtained from WGCNA (red, positive correlation; blue, negative correlation). (E) Top 10 pathways determined by KEGG enrichment analysis of genes from ME2.

and DFI of OV patients; *Pseudomonas* sp. B14-6, *Klebsiella michiganensis*, *Buchnera aphidicola*, *Paraburkholderia edwinii*, *Comamonas aquatica*, *Veillonella nakazawae*, *Corynebacterium jeikeium,* and *Bifidobacterium subtile* especially were significantly associated with shorter OS, DSS, PFS, and DFI of patients, while *Gemmatirosa kalamazoonesis* was significantly associated with longer OS, DSS, PFS, and DFI (Fig. S4 to S7).

In summary, the intratumoral microbiome profile was significantly associated with the prognosis of patients with OV, and the prediction model consisting of 32 microbes showed great prognostic value for OV patients.

**Integrated analysis of multiomics reveals potential host-immune interactions.** We further explored the host's immune-microbe interactions within the TME. The correlation network (12 nodes, 9 edges) between 32 prognostic microbes and 7 immune cells that were significantly different between clust1 and clust2 demonstrated that 5 types of immune cells were significantly associated with 7 species ($P < 0.05$) (Fig. 5A). M1 was identified as the hub, associated with 5 species. M1 was enriched in clust2 and positively associated with two protective species, *Achromobacter deleyi* and *Microcella alkaliphila*, while negatively associated with three risk species, *Devosia* sp. LEGU1, *Ancylobacter pratisalsi,* and *Acinetobacter*

*seifertii*. Differential analysis between high- and low-risk groups showed that the high-risk group had a higher proportion of M1 than the low-risk group (Fig. 5B). There was a weak negative association between M1 and risk score ($r = -0.146$, $P = 0.021$) (Fig. 5B). Survival analysis showed that the survival time in the high-M1 group, including OS, DSS, and DFI, was longer than that in the low-M1 group (Fig. 5C).

Next, we identified 28 gene modules (MEs) using weighted gene correlation network analysis (WGCNA) based on the immune gene expression data, and 17 MEs enriched for at least 1 KEGG pathway were used for downstream analysis (Fig. S8). We analyzed the relationship between 17 MEs and 7 prognostic species associated with immune cells. Spearman rank correlation network analysis (15 nodes, 21 edges) (Fig. 5D) demonstrated that 10 MEs (58.8%) were significantly associated with 5 species ($P < 0.05$), among which 3 species (*Achromobacter deleyi*, *Corynebacterium jeikeium,* and *Acinetobacter seifertii*) and 3 MEs (ME2, ME14, and ME23) were hubs (degree $\geq$ 3). Genes in ME2 were enriched in KEGG pathways related to some nervous system diseases and reactive oxygen species, which is a kind of chemical carcinogen (Fig. 5E); genes in ME14 and ME23 were enriched in the spliceosome pathway and the herpes simplex virus 1 (HSV-1) infection pathway, respectively (Fig. S8).

**Cell experiments demonstrated the inhibitory effect of *Acinetobacter seifertii* on macrophage migration.** To demonstrate whether *Acinetobacter seifertii* can affect macrophage polarization, we treated peritoneal macrophages with *Acinetobacter seifertii* (As-10 group), *Acinetobacter seifertii*-treated ID8 cells (ID8-As-10-Cell group), conditioned medium (CM) of *Acinetobacter seifertii*-treated ID8 cells (ID8-As-10-CM group), untreated ID8 cells (ID8-Cell group), CM of untreated ID8 cells (ID8-CM group), and Dulbecco's modified Eagle medium (DMEM), respectively. Compared to untreated ID8 cells, the expression of *TNFA* and *iNOS*, which are markers of M1 macrophages, was greatly increased in groups treated with *Acinetobacter seifertii* (Fig. 6A).

We further performed Transwell experiments to investigate whether *Acinetobacter seifertii* could inhibit macrophage migration. The results showed that the migration of macrophages was obviously decreased in groups treated with *Acinetobacter seifertii*, especially the ID8-As-10-Cell group (Fig. 6B and C). These results determined that the negative correlation between *Acinetobacter seifertii* and M1 cells is derived from the inhibitory effect of this bacterium on macrophage migration.

Together, the findings of our study showed that the tumor immune microenvironment was closely related to the intratumoral microbiome. Totals of 58.8% (15 out of 17) MEs and 71.4% (5 out 7) immune cells were significantly associated with prognostic microbes. Especially, M1 was strongly associated with 5 species, and there was a negative correlation between *Acinetobacter seifertii* and M1 cells.

## DISCUSSION

Recent studies have determined the existence of intratumoral microbiota in a variety of nongastrointestinal tumors, including OV (8). However, the role of intratumoral microbiota in the development and prognosis of OV remains largely unknown. In this study, OV could be classified into two subtypes, namely, immune-enriched and immune-deficient subtypes, according to TME characteristics. Moreover, the intratumor microbial profiles were found to be different between these two subtypes, which may contribute to the prognosis of patients with OV.

The diversity of TME characteristics in patients with OV has been highlighted in previous reports, proving that it could serve as a basis for OV classification (4, 5, 7, 19). OV could be classified into four subtypes, including immunoreactive, mesenchymal, proliferative, and differentiated types (19). Bagaev et al. (4) have also implemented a pancancer classification identifying four TME subtypes termed immune enriched and fibrotic, immune enriched and nonfibrotic, fibrotic, and immune depleted. It is worth noting that immune-enriched and immune-deficient subtypes in this study showed strong concordance with the molecular subtypes mentioned above, indicating the robustness of TME subtypes. Patients in the immune-enriched subtype group (clust2) were found to have higher stromal and immune scores, with a higher proportion of CD8$^+$ T cells and the M1 type of macrophages (M1), and also have a better survival prognosis than patients in the immune-deficient subtype

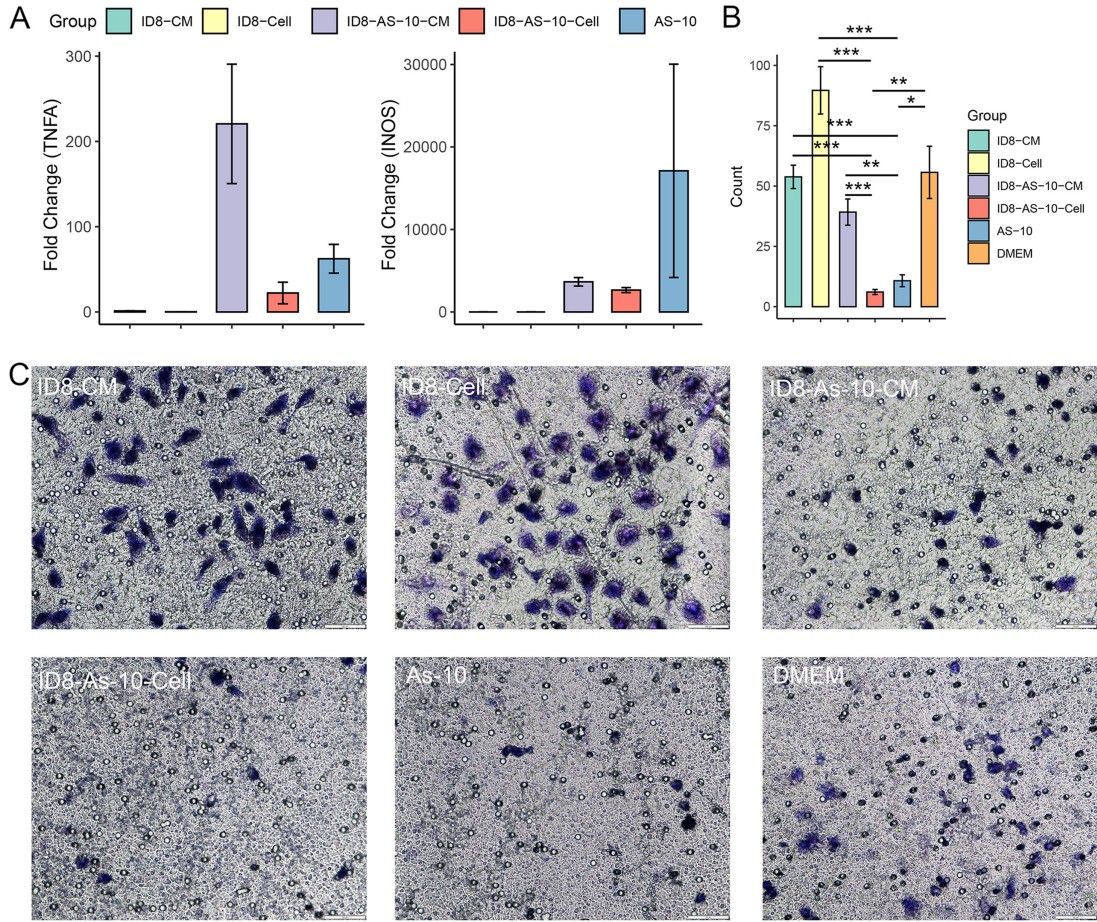

**FIG 6** Cell experiments demonstrating the inhibitory effect of *Acinetobacter seifertii* on macrophage migration. (A) Fold changes in expression levels of *TNFA* and *iNOS* compared to those of the DMEM group. Mean and SEM are shown. (B) Number of migrated macrophages of five fields taken at a magnification of ×400 per group. Mean and SEM are shown. (*, $P < 0.05$; **, $P < 0.01$; ***, $P < 0.001$). (C) Transwell migration assays of macrophages taken at a magnification of ×400 per group.

group (clust1). M1 macrophages are potent immune effector cells, which could upregulate genes involved in antigen processing and presentation as well as enhance T-cell responses by secreting costimulatory molecules (20–22). Considerable evidence has suggested that the infiltration of cytotoxic CD8+ T lymphocytes in OV is significantly associated with a better prognosis (3–5, 7, 23). These results suggested that CD8+ T cells and M1 macrophages are the key TME-related cells that play roles in regulating the survival of OV patients with immune-enriched and immune-deficient subtypes. Gene mutations occurring in the coding region during carcinogenesis can produce proteins that are not found in normal cells. These proteins, called neoantigens, are underlying factors determining tumor immunogenicity (24). Concordantly, our research showed that the immune-enriched subtype exhibited higher TMB and more potential neoantigens.

Since the application of neoantigens in therapeutic cancer vaccines has been widely discussed (25, 26), we here focused on the intratumoral microbiota, which is also an important component of TME and could cause inflammation or local immune suppression (27–33). We found that the intratumoral microbiome profiles were significantly different between clust1 and clust2. *Acinetobacter baumannii* is an opportunistic pathogen significantly enriched in clust2. It can cause severe hospital-acquired infections (34), which may contribute to generating a proinflammatory microenvironment. Besides, it has been proven that *Fusobacterium nucleatum* can migrate from the oral cavity to the colorectum and cause numerous diseases, including colorectal cancer (35). It can not only amplify tumorigenesis through the FadA adhesin and outer membrane vesicle (28, 31, 36–41) but also protect tumors from immune cell attack by binding to human inhibitory receptor. Consistently, *Fusobacterium nucleatum*

was significantly enriched in clust1 patients with an immune-deficient phenotype and associated with shorter DFI and PFS. There were also intricate immune-microbe interactions driven by a variety of contact-independent mechanisms.

Secreted metabolites and catabolites from microbes are active across different tumorous niches (11, 42, 43). We identified cholesterol metabolism (map04979) and taurine and hypotaurine metabolism (map00430) as differential KEGG pathways with higher abundance in clust1. In line with our results, it has been reported that gut microbes can metabolize cholesterol to produce 27-hydroxycholesterol, and the latter had been reported to promote the growth and metastasis of estrogen receptor-positive breast cancer (44, 45); taurine could be converted to hydrogen sulfide, which is a genotoxin that initiates colon cancer (46). The intratumoral microbiome may also play a part in carcinogenesis and prognosis indirectly via metabolites. In addition, we found that the PI3K/Akt signaling pathway and the phosphatidylinositol signaling system were more abundant in clust1. The activation of such pathways has been reported to result in an increase in cell proliferation, migration, invasion, and chemotherapy resistance in OV (47), suggesting that environmental information processing of intratumoral microbes may also program antitumor immunity and the outcome of OV patients.

Anticancer therapies have demonstrated strong links between distinct commensals and protective antitumor immune responses (11, 33, 43, 48, 49), in which innate immunity plays a critical role. Macrophages are the parts of antitumorigenic mononuclear phagocytes that are involved in the detection, phagocytosis, and destruction of bacteria and other harmful organisms. They could link innate and adaptive immunity through antigen presentation. According to activation state and functions, macrophages can be classified as M1 and M2, which are a "double-edged sword" in antitumor immunity (50). M1 participates in the positive immune response and functions as an immune monitor and thus plays a critical role in stopping the spread of cancer cells (43). Lam et al. have determined that microbiota from immune checkpoint blockade responders could shape the mononuclear phagocyte landscape in the TME (43). Lipopolysaccharide, which is an essential component of the outer membrane of all Gram-negative bacteria, and Toll-like receptors are stimulators to polarize macrophages into M1 (51). In our research, the proportion of M1 in tumors was significantly associated with longer OS, DSS, and PFS and positively associated with two protective Gram-negative bacteria, *Achromobacter deleyi* and *Microcella alkaliphila*, suggesting that lipopolysaccharide from Gram-negative bacteria may be important for promoting antitumor immunity. Besides, it has been reported that the microbiome in primary tumor tissue exerts potent suppressive influences on the inflammatory TME by modulating the activity of the tumor-associated macrophages (52, 53). Our research showed that *Acinetobacter seifertii*, a risk species negatively associated with M1, has an inhibitory effect on macrophage migration, indicating a potential mechanism by which the intratumoral microbiota affects the TME. Further studies are needed to determine the inhibition mechanism of *Acinetobacter seifertii*.

Our research also found that ME23 enriched in the HSV-1 infection pathway was positively associated with M1. HSV-1 is the cause of oral herpes and genital herpes, and M1 plays a central role in acute and chronic HSV-1 infection (54, 55). Furthermore, HSV has been considered and developed as an oncolytic virus for cancer therapy for 30 years (56). The potential role of HSV-1 infection in the progression and prognosis of OV is noteworthy.

Previous studies support the contribution of the intratumoral microbiota to cancer progression and treatment prognosis (29, 33, 48, 49, 57–59). Riquelme et al. demonstrated that the composition of the pancreatic adenocarcinoma microbiome could influence the host immune response and natural history of the disease (33). In this study, we constructed a Cox model consisting of 32 microbes to investigate the relationship between the intratumoral microbiota and prognosis for OV patients and found that the model has a strong ability to distinguish the patients into high- and low-risk groups. Patients in the low-risk group had significantly longer OS and DSS, suggesting that tumor microbiome sequencing could be used to stratify patients for adjuvant trials and that microbiome interventions may aid antitumor therapy.

There were several limitations deserving further analysis and discussion. A major limitation was that our research preliminarily explored the relationship between intratumoral microbes and the TME and prognosis of OV, while the causal relationship and specific mechanism require more rigorous experiments for validation. Another limitation was that we generated the results by using the Kraken2 pipeline, which obtains microbiome information from RNA sequencing data. Therefore, it is necessary to verify the results by metagenome sequencing and/or PCR analysis.

## MATERIALS AND METHODS

**Data accession.** For 373 patients with OV in TCGA, the clinical data, survival data, and both bam files and read counts of RNA sequencing data were downloaded for this study. Briefly, the raw bam files of RNA sequencing data were obtained from the NCI Genomic Data Commons (GDC) via the GDC's application programming interface (API) in accordance with the TCGA Data Use Certification Agreement; clinical data, raw read counts, and fragments per kilobase million (FPKM) data of the RNA sequencing data were downloaded using the TCGAbiolinks R package (v2.18.0) (60); transcripts per million (TPM) data and logTPM data of RNA sequencing data were transformed from FPKM data; and the survival data were obtained from the TCGA Pan-Cancer Clinical Data Resource (TCGA-CDR) (61). To describe the immune landscape of patients with OV in TCGA, the following data were collected. Twenty-nine knowledge-based functional gene expression signatures (Fges) representing the major functional components and immune, stromal, and other cellular populations of the tumor signatures were acquired from Bagaev et al. (4). TCR/BCR diversity, tumor purity, and neoantigen load data were obtained from the pancancer immune landscape project conducted by Thorsson et al. (3). Immunologic signature gene sets representing cell states and perturbations within the immune system were obtained from Molecular Signatures Database (MSigDB v7.5.1) (62).

**Metagenomic profiling based on RNA sequencing data of OV.** Based on mapping information of raw BAM files, sequencing reads that did not align with Genome Reference Consortium Human Build 38 (hg38) were retained to identify microbial reads. After quality assessment using FastQC (v0.11.8) and MultiQC (v1.9), the reads were filtered using Trimmomatic (v0.39) with the following options: ILLUMINACLIP: TruSeq2-PE.fa:2:30:10 SLIDINGWINDOW:4:20 MINLEN:35. By using Kraken2 (v2.1.2), the obtained clean reads were aligned against the default standard database to identify bacterial and archaeal reads. Bracken (v2.5.0) was used to quantify taxon abundance from Kraken2 profiles, and microbial count data were obtained. Furthermore, reads of bacteria and archaea were assembled into contigs using MEGAHIT (v1.2.9). Prodigal (v2.6.3) was used to predict protein coding genes for contigs with the "meta" option. CD-HIT (v4.8.1) was used to build the nonredundant gene sets. Salmon (v1.8.0) was used to quantify the gene abundance in each sample. Finally, the KEGG functional annotation of nonredundant gene sets were applied using eggNOG-Mapper (v2.1.6) and diamond (v2.0.11) based on the eggNOG database (v5.0.2).

**Microbial data decontamination and normalization.** To mitigate external contamination as much as possible, we applied the decontamination pipeline described by Poore et al. (9). We first used the associated decontam R package (63) with the recommended hyperparameter threshold ($P = 0.1$) to conduct the filtering. We processed the data in batches corresponding to the sequencing plate, whereby species ($n = 16$) identified as a contaminant were subsequently discarded before downstream analysis. Next, we removed genera typically found in "negative blank" reagents (1,168 species from 67 genera) described by Salter et al. (64) (stringent decontamination, 70% of microorganisms were considered to be contaminated). Manual literature inspection reallowed 654 species from 11 genera that were potentially pathogens or commensals (likely decontamination, 17% of microorganisms were discarded as contaminated). Downstream analysis was primarily based on likely decontamination data sets.

To create an approximately normally distributed, we transformed count data to counts per million (CPM) data using the NormalizeData function of the Seurat R package (v4.1.0) (65). We add 1 to the CPM data and performed log transformation to obtain logCPM data.

**Immune landscape and classification of OV.** First, we performed gene set variation analysis (GSVA) in the GSVA R package (v1.38.2) (66) to estimate variation of the 29 Fges in each sample base on logTPM data. Then, based on the matrix derived from GSVA, we applied the k-means function in the R Stats package (v4.0.3) to classify OV patients into two clusters (clust1 and clust2) using the k-means clustering method.

To describe the immune landscapes of these two subtypes, we performed the following analysis. Cell type identification by estimating relative subsets of RNA transcripts (CIBERSORT) (67) is a method for characterizing the cell composition of complex tissues using their gene expression profiles. We first estimated the proportions of 22 types of infiltrating immune cells based on TPM data using CIBERSORT R script (v1.03). The estimation of stromal and immune cells in malignant tumor tissues using expression data (ESTIMATE) (68) is a method that uses gene expression signatures to infer the fraction of stromal and immune cells in tumor samples. We used the estimate R package (v1.0.13) to calculate the overall stromal and immune scores in cancer. Furthermore, the differentially expressed genes between two immune subtypes of OV patients were identified using DESeq2 R package (v1.30.1) (69). Finally, function analysis, including WGCNA and KEGG enrichment analysis, were conducted using the WGCNA R package (v1.70-3) (70) and clusterProfiler R package (v3.18.1) (71), respectively.

**Intratumoral microbial analysis of OV.** First, the alpha-diversity (measured by observed and Shannon indexes) and beta-diversity (measured by Bray-Curtis distance) were analyzed and compared in two immune subtypes using MicrobiotaProcess (v1.2.2) (72) and vegan (v2.8-7) R package (73), respectively. Then, LEfSe (74) analysis (http://huttenhower.sph.harvard.edu/galaxy) was used to identify differential taxa between the two immune subtypes identified in this study. Finally, the microbial cooccurrence network analysis was

performed using the CCLasso algorithm to elucidate microbiota interactions by MetagenoNets (75) (https://web.rniapps.net/metagenonets/).

**Survival analysis.** We aimed to identify microbe signatures associated with prognosis, including OS, DSS, PFS, and DFI, of OV. Based on the logCPM normalized data, the univariate Cox regression analyses (adjusted for age) were conducted to identify potential prognostic species using the survival R package (v3.2-11). For OS, species with a $P$ of $<0.05$ served as potential prognostic signatures for further selection. We next divided all patients into training and validation sets 8:2. In the training set, LASSO-penalized Cox regression analysis was performed to screen prognostic species using the glmnet R package (v4.1-2) (76), which could avoid overfitting and reduce multicollinearity. Tenfold cross-validations were performed to define the optimal value of the lambda penalty parameter; this resulted in the weight of most of the potential prognostic species decreasing to zero, and a relatively small number of prognostic microbes with a weight of nonzero remained. Lastly, multivariate Cox regression analysis was used to construct a microbe-based prediction model with the rms R package (v6.2-0) (77). The C-index is used to assess the model discrimination power that can taken into account censored data. The risk score can be estimated from the Cox model as follows:

$$\text{risk score} = \exp\left[\sum_{i=1}^{p} b_i x_i - \sum_{i=1}^{p} b_i \overline{x}_i\right]$$

where the coefficients ($b_1, b_2, \ldots, b_p$) measure the impact (i.e., the effect size) of covariates, $x_i$ is the value of the $i$th covariate from the subjects, and $\overline{x}_i$ is the mean value of the $i$th covariate. The lower quartile of risk score was used as the cutoff for classifying patients into low- and high-risk score groups.

Kaplan-Meier curves plotted by the survminer R package (v0.4.9) (78) were used to (i) compare survival prognoses between the two subtypes identified in this research, (ii) assess the prognostic capacity of the prognostic species, and (iii) compare survival prognoses between high-M1 and low-M1 groups.

**Joint analysis of immune characteristics and intratumoral microbiome.** To explore the interactions between host's immune characteristics and intratumoral microbes, we constructed three correlation networks consisting of infiltrating immune cells and intratumoral microbes, MEs and intratumoral microbes, and MEs and immune cells. Spearman rank correlation coefficients and $P$ values were calculated using the corr.test function in the psych R package (v2.1.9) (79). After filtering of correlations with $P < 0.05$, Cytoscape (v3.9.0) (80) was used to visualize the correlation networks.

**Statistical analysis.** Differences in continuous variables such as microbial diversity and immune indices were addressed using nonparametric Wilcoxon rank sum testing. and chi-square tests were used to analyze the categorical variables. The Benjamini-Hochberg false discovery rate test was used to correct for multiple comparisons. Principal coordinate analysis (PCoA) was performed on the Bray-Curtis distance of Hellinger transformed microbial data using the vegdist function of the vegan R package (v2.5-7). Permutational multivariate analysis of variance (PERMANOVA) was performed to compare the beta-diversities (Bray-Curtis distance matrix) between the two clusters identified by this study. $R^2$ is the coefficient of determination, which represents the percentage of the total variance of the distance matrix explained by the tumor immune condition. The $P$ value is the possibility that the distribution will be the same between groups. The smaller the $P$ value, the smaller the probability of the same distribution. We analyzed the similarities to test whether there is a significant difference between two groups of sampling units by using analysis of similarities (ANOSIM). The $R$ value is the difference between the means of the distance ranks between groups and within groups. The $P$ value is used to indicate whether there is a significant difference between groups. If $P$ is $<0.05$, the dissimilarities within and between groups were considered to be significantly different. Baseline data were compared between groups using the compareGroups R package (v4.5.1) (81). A linear regression model between the proportion of M1 and the risk score calculated using the Cox model was constructed using the lm function in the stats R package (v4.0.3).

**Cell line.** The ID8 cell line was obtained from Millipore and cultured in DMEM supplemented with 10% fetal bovine serum (FBS) and 1% penicillin-streptomycin. The ID8 cell line was cultured at 37°C and 5% $CO_2$.

**Isolation of peritoneal macrophages.** Macrophages were induced by injecting 1 mL of sterile 6% soluble starch solution into the peritoneal cavity of wild-type (WT) C57BL/6 male mice at 6 weeks age. Three days later, peritoneal exudate cells were isolated by injecting 10 mL of sterile DMEM basal culture into the peritoneal cavity and slowly withdrawing the fluid. The isolated peritoneal macrophages were cultured in DMEM supplemented with 10% FBS and 1% penicillin-streptomycin (complete DMEM) at 37°C and 5% $CO_2$.

**Culture of *Acinetobacter seifertii*.** *Acinetobacter seifertii* A354 was obtained from the YYYS lab, Zhejiang University. Under sterile conditions, bacterial liquid suspension was cultured in liquid culture medium (tryptic soy broth) and enriched at 37°C for 24 h under aerobic conditions. After measurement of optical density at 600 nm ($OD_{600}$) and counting bacterial CFU, the bacterial standard curve was obtained to calculate the bacterial fluid volume needed for coculture experiments.

**Culture of macrophages.** Peritoneal macrophages were cultured using the following media. To analyze the combined effect of *Acinetobacter seifertii* A354 and ID8 cells on macrophages, the ID8 cell line was treated with *Acinetobacter seifertii* A354 (multiplicity of infection [MOI] = 10) when the cells reached 70 to 80% confluence. After 6 h, the cell conditional medium (CM) was collected and filtered using a 0.45-$\mu$m filter membrane to remove residual bacteria and cells (ID8-As-10-CM group), while the ID8 cells were digested and made into cell suspension (ID8-As-10-Cell group). The CM and the cell suspension of ID8 cells were used as reagent control groups, namely, ID8-CM and ID8-Cell groups, respectively. To analyze the effect of *Acinetobacter seifertii* A354 on macrophages, *Acinetobacter seifertii* A354 was diluted with complete DMEM (As-10 group), while complete DMEM was used for the blank control group.

**RNA extraction and reverse transcription qPCR.** Total mRNA from macrophages was extracted with a FastPure cell/tissue total RNA isolation kit V2 (Vazyme). mRNA was reverse transcribed using the HiScript

III RT supermix for quantitative PCR (qPCR) (+gDNA wiper) (Vazyme). qPCR analyses were performed using ChamQ universal SYBR qPCR master mix (Vazyme). mRNA expression was assessed by the comparative cycle threshold ($C_T$) method ($2^{-\Delta\Delta CT}$); beta-actin genes (forward primer: 5′-GGCTGTATTCCCCTCCATCG-3′, revers primer: 5′-CCAGTTGGTAACAATGCCATGT-3′) were used as housekeeping genes. TNFA genes (forward primer: 5′-CCCTCACACTCAGATCATCTTCT-3′, revers primer: 5′-GCTACGACGTGGGCTACAG-3′) and iNOS genes (forward primer: 5′-GTTCTCAGCCCAACAATACAAGA-3′, revers primer: 5′-GTGGACGGGTCGATGTCAC-3′) are used as markers of M1 macrophages.

**Macrophage migration test.** Transwell chambers (8.0 $\mu$m; Corning) without Matrigel coating (Corning) were used for cell migration analysis. Macrophage suspensions were diluted to $4 \times 10^5$ cells/mL using serum-free DMEM and transferred into the upper compartment, with the culture media mentioned above added to the lower compartments. Each chamber was seeded with the 100-$\mu$L macrophage suspension. Before staining and counting, migrated cells were incubated for 48 h at 37℃ and 5% $CO_2$. The average numbers of migrating or invading cells in each group were calculated in at least five fields.

**Data availability.** The data sets supporting the conclusions of this article are available in the Database of Genotypes and Phenotypes (dbGaP) repository for TCGA (accession no. phs000178.v11.p8).

## SUPPLEMENTAL MATERIAL

Supplemental material is available online only.

**SUPPLEMENTAL FILE 1**, XLS file, 0.2 MB.
**SUPPLEMENTAL FILE 2**, DOCX file, 0.02 MB.
**SUPPLEMENTAL FILE 3**, PDF file, 4.1 MB.

## ACKNOWLEDGMENTS

We thank Yunsong Yu at the Department of Infectious Diseases, Sir Run Run Shaw Hospital, College of Medicine, Zhejiang University, for donating *Acinetobacter seifertii* A354.

We have no conflicts of interest to declare.

This study was supported by National Natural Science Foundation of China grant no. 82172320, TaiShan Industrial Experts Program grant no. tscy20190612, and the Shandong University Outstanding Young Scholars Program.

Lei Zhang and Chuandi Jin conceived and designed the project. Each author contributed significantly to the submitted work. Dashuang Sheng and Kaile Yue collected and organized all data, performed data analysis and cell experiments, and drafted the manuscript. Lanlan Zhao sorted out the analytical methods. Lei Zhang, Chuandi Jin, and Guoping Zhao revised the manuscript. All authors read and approved the final manuscript.

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
