## [Reviewer comments · Microbiology Spectrum]

Microbiology Spectrum

The interaction between intratumoral microbiome and immunity is related to the prognosis of ovarian cancer

Lei Zhang, Dashuang Sheng, Kaile Yue, Lanlan Zhao, Chuandi Jin, and Guoping Zhao

Corresponding Author(s): Lei Zhang, Shandong University

Review Timeline:

Submission Date:	September 3, 2022
Editorial Decision:	November 4, 2022
Revision Received:	January 17, 2023
Accepted:	February 22, 2023

Editor: Se-Ran Jun

Reviewer(s): The reviewers have opted to remain anonymous.

Transaction Report:

DOI: <https://doi.org/10.1128/spectrum.03549-22>

November 4, 2022

Prof. Lei Zhang
Shandong University
Jinan
China

Re: Spectrum03549-22 (The interaction between intratumoral microbiome and immunity is related to the prognosis of ovarian cancer)

Dear Prof. Lei Zhang:

Link Not Available

Sincerely,

Se-Ran Jun

Journals Department
Reviewer comments:

Reviewer #1 (Comments for the Author):

The experimental design is good, the results are interesting, and the writing is smooth and easy to follow. However the study is totally computational without any attempt to validate by wet lab experiments. I would suggest at least a cell-line based assay to prove for example the immunomodulatory effect of some identified microbial species-or their effect on reactive oxygen species, etc.

Reviewer #2 (Comments for the Author):

Tumors are always associated with microbial communities, which may play crucial roles in the modulation of oncogenesis. However, how the enclosed or peripheral microbiome interacts with the tumor microenvironment and affects the prognosis of the tumor remains elusive. In this work, Sheng et al. reanalyzed the RNA-seq data of TCGA-OV and extracted the non-human reads for tumor-associated microbiota and functional profiling. Integrating the immune gene expression, the authors explored whether there were relationships between the OV tumor microbiota and tumor subtypes (immune-deficient and immune-enriched). By leveraging the survival data, they tried to identify potential prognosis biomarker bacterial species. The work did a comprehensive analysis of OV tumor microbiota and a close investigation of the relationships with tumor microenvironments and prognosis, and thus will be of interest to a broad range of readers. I have some minor comments.

Minor comments:

1. What's the difference between the MFP TCGA and the Fges-based cancer microenvironment subtypes? Why choose the Fges-based method to differentiate the specimen groups? Based on the demonstration in the introduction, the Fges-based method classified ovarian cancer (OV) into four subtypes, i.e. (1) the immune-enriched, fibrotic subtype; (2) the immune-enriched, non-fibrotic subtype; (3) the fibrotic subtype and (4) the immune-deficient subtype. In the results session, only two subtypes (immune-enriched and immune-deficient subtypes) were designated. Why? Did the immune-enriched subtype include both fibrotic and non-fibrotic groups?
2. Whether the transcriptome sequencing processed a poly A filtration, and if so, how much would this affect the bacterial profiling results?
3. "Fig. 1a" in the main text should be capitalized.
4. Fig. 2: Did the authors measure the absolute bacterial cell numbers? Were there differences between the two OV subtypes?
5. Line 118-119: "the beta-diversity analysis showed that intratumoral microbial profiles were different between clust1 and clust2 (P=0.02, Fig. 3a)". As shown in Fig. 3A, only coordinate 1 had slightly and significant variations, with a df approximately approaching 1. This sentence should be rephrased. Line 119-120, "it was more dissimilar among individuals in the clust1". How did the comparison was processed? Was the dissimilarity among all three groups? More dissimilar among individuals in the clust1 compared to both or which group (between or the clust2)? This should be clarified.
6. Line 142-144, "a prediction model consisted of 6 favorable species, 26 risk species, and age at diagnosis was constructed". The abundance of 193 bacterial species was associated with OS. How did the 6 favorable and 26 risk species were picked?

Staff Comments:

Preparing Revision Guidelines

Please return the manuscript within 60 days; if you cannot complete the modification within this time period, please contact me. If you do not wish to modify the manuscript and prefer to submit it to another journal, please notify me of your decision immediately so that the manuscript may be formally withdrawn from consideration by Microbiology Spectrum.

Reviewer comments:

Reviewer #1 (Comments for the Author):

The experimental design is good, the results are interesting, and the writing is smooth and easy to follow. However, the study is totally computational without any attempt to validate by wet lab experiments. I would suggest at least a cell-line based assay to prove for example the immunomodulatory effect of some identified microbial species-or their effect on reactive oxygen species, etc.

Response: We appreciate the kind comments and thoughtful suggestions. In this study, a total of 32 microbes with prognostic value were selected using Lasso-penalized Cox regression analysis, and five of them including *Acinetobacter seifertii* were significantly associated with M1 macrophages. As previous studies have reported that tumor-infiltrating M1 macrophage is a protective factor for OV patients, we performed two cellular experiments to validate the inhibitory effect of *Acinetobacter seifertii* on macrophages.

1) To demonstrate whether *Acinetobacter seifertii* can affect macrophage polarization, we treated peritoneal macrophages with *Acinetobacter seifertii* (MOI=10), *Acinetobacter seifertii*-treated ID8 cells (MOI=10), conditioned medium (CM) of *Acinetobacter seifertii*-treated ID8 cells (MOI=10), untreated ID8 cells, CM of untreated ID8 cells, and DMEM medium, respectively. Compared to untreated ID8 cells, the expression levels of *TNFA* and *iNOS*, which are markers of M1 macrophages[1], were greatly increased in groups treated with *Acinetobacter seifertii*, suggesting that such bacteria can induce the M1 polarization of macrophages (Fig. 6A).

2) We further performed experiments to investigate whether this bacterium could inhibit migration of macrophages. The results of transwell experiments showed that the migration of macrophage was decreased obviously in groups treated with *Acinetobacter seifertii*, especially macrophages co-cultured with *Acinetobacter seifertii*-treated ID8 cells (Fig. 6B, C). These results demonstrated that the negative correlation between *Acinetobacter seifertii* and M1 cells is derived from the inhibitory effect of this bacterium on macrophage migration.

Taken together, by following the reviewer's suggestion, these experimental results provide support for our data analysis to a certain extent. We have added the contents of this part to the methods and results of the revised manuscript and discussed this section as well. (Line 35, 48, 87-88, 178-191, 194-195, 263-268, 407-447)

Figure 6

Figure 6. Cell experiments demonstrated the inhibitory effect of *Acinetobacter seifertii* on macrophage migration. (A) The fold changes of expression levels of *TNFA* and *iNOS* compared to DMEM group. Mean and SEM are shown. (B) Number of migrated macrophages of five fields taken at a magnification of $\times 400$ per group. Mean and SEM are shown. (C) Transwell migration assays of macrophages taken at a magnification of $\times 400$ per group. * $P < 0.05$; ** $P < 0.01$; *** $P < 0.001$

Reviewer #2 (Comments for the Author):

Tumors are always associated with microbial communities, which may play crucial roles in the modulation of oncogenesis. However, how the enclosed or peripheral microbiome interacts with the tumor microenvironment and affects the prognosis of the tumor remains elusive. In this work, Sheng et al. reanalyzed the RNA-seq data of TCGA-OV and extracted the non-human reads for tumor-associated microbiota and functional profiling. Integrating the immune gene expression, the authors explored whether there were relationships between the OV tumor microbiota and tumor subtypes (immune-deficient and immune-enriched). By leveraging the survival data, they tried to identify potential prognosis biomarker bacterial species. The work did a comprehensive analysis of OV tumor microbiota and a close investigation of the relationships with tumor microenvironments and prognosis, and thus will be of interest to a broad range of readers. I have some minor comments.

Minor comments:

1. What's the difference between the **MFP TCGA** and the Fges-based cancer microenvironment subtypes? Why choose the Fges-based method to differentiate the specimen groups? Based on the demonstration in the introduction, the Fges-based method classified ovarian cancer (OV) into four subtypes, i.e. (1) the immune-enriched, fibrotic subtype; (2) the immune-enriched, non-fibrotic subtype; (3) the fibrotic subtype and (4) the immune-deficient subtype. In the results session, only two subtypes (immune-enriched and immune-deficient subtypes) were designated. Why? Did the immune-enriched subtype include both fibrotic and non-fibrotic groups?

Response: Thank you for the opportunity to clarify this point. We chose the binary classification instead of the four classification for downstream analysis for the following reasons.

1) Immune cells such as macrophages, lymphocytes, and natural killer cells are the important components of the tumor immune microenvironment. Early infiltration of these cells into tumor is crucial for tumor control. Previous studies have reported that tumor infiltration of M1 macrophages and CD8⁺ T cells are important for the prognosis of ovarian cancer treatment[2, 3]. As the relationship between intratumoral microbes and intratumoral immune infiltration remains unclear, we mainly focused on the relationship between tumor immune infiltration and intratumoral microbiota. According to the variation scores of 29 Fges obtained from Gene Set Variation Analysis (GSVA), the ovarian cancer samples were re-grouped by K-means clustering into immune-enriched and immune-deficient types. As shown in the Extended Fig. 1A, the immune-enriched subtype includes most fibrotic and non-fibrotic groups.

2) We have performed comparison among the four subtypes and found significant differences in microbial composition (Extended Fig. 1B, C). Besides, the survival outcome was the primary outcome of interest in our study. The survival analysis showed that the difference in overall survival among four subtypes was not as obvious as that between two subtypes (Extended Fig. 1D, E).

Hence, we performed our analysis based on two subtypes (immune-enriched and immune-deficient subtypes). We have revised the contents of this part to the introduction of the manuscript. (Line 60-66)

Extended Figure. 1 The association and difference between binary and four classification of TME. (A) Percentages of OV subtypes identified by Bagaev et al. across clust1 and clust2. (B) Boxplots of alpha-diversity compared among the 4 TME subtypes taken from Bagaev et al. (C) PCoA and boxplot shown along the first two principal coordinates on the Bray-curtis distances. Ellipses represent the 95% confidence interval around the group centroid. *P* value were determined by PERMANOVA. (D-E) Kaplan-Meier (K-M) curves of OS in OV patients stratified by TME subtype classification taken from Bagaev et al.(D) and this study (E). The *P* value was calculated with the log-rank test.

2. Whether the transcriptome sequencing processed a poly A filtration, and if so, how much would this affect the bacterial profiling results?

Response: Thanks for your comments. The samples from TCGA-OV were performed whole-transcriptome sequencing and the total RNA were extracted using the DNA/RNA AllPrep kit (QIAGEN) with ribosomal depletion[4-6].

3. "Fig. 1a" in the main text should be capitalized.

Response: We appreciate your kind suggestion and we have amended the manuscript

accordingly.

4. Fig. 2: Did the authors measure the absolute bacterial cell numbers? Were there differences between the two OV subtypes?

Response: Thank you very much for pointing this out. Due to the limitations of the sequencing protocol, the absolute bacterial cell numbers were not measured. Instead, we calculated the ratio of microbial counts to the number of RNA-seq reads to approximate the microbial load in the tumors and found that there was no significant difference in bacterial load between the two groups.

5. Line 118-119: "the beta-diversity analysis showed that intratumoral microbial profiles were different between clust1 and clust2 ($P=0.02$, Fig. 3a)". As shown in Fig. 3A, only coordinate 1 had slightly and significant variations, with a df approximately approaching 1. This sentence should be rephrased. Line 119-120, "it was more dissimilar among individuals in the clust1". How did the comparison was processed? Was the dissimilarity among all three groups? More dissimilar among individuals in the clust1 compared to both or which group (between or the clust2)? This should be clarified.

Response: With our apologies for the lack of clarity, permutational multivariate analysis of variance (PERMANOVA) is the analysis of variance using distance matrices for partitioning distance matrices among sources of variation and fitting linear models (e.g., factors, polynomial regression) to distance matrices. Statistical inferences are made in a distribution-free setting using permutational algorithms. As shown in Fig. 3A,

PERMANOVA was performed to compare the β -diversity (distance matrix) between the 2 clusters identified by this study. R^2 is the coefficient of determination, which represents the percentage of the total variance of the distance matrix explained by the tumor immune condition. The P value is the possibility that the distribution will be the same between the groups. The smaller the P value, the smaller the probability of the same distribution. When $P < 0.05$, the microbial distribution is considered to be significantly different between groups. Also, comparison of coordinate 1 and coordinate 2 between two cluster was calculated using Mann-Whitney U test and displayed in Figure 3A. However, the PERMANOVA test was a more comprehensive

method for testing the difference between two distance matrices. Therefore, the β -diversity were different between clust1 and clust2 based on these above results. We have added the details for PERMANOVA test in the revised manuscript. (Line 394-399)

For line 119-120, we analyze the similarities to test whether there is a significant difference between two or more groups of sampling units. The procedures of Analysis of Similarities (ANOSIM) are 1) calculating the distance between pairs of samples; 2) ordering the distance between pairs of samples from large to small and calculating the rank, and 3) calculating the difference between the mean of the distance rank between groups and the mean of the distance rank within groups, namely R . If $R > 0$, the within-group distance is smaller than the between-group distance, that is, the grouping is effective, which is similar to the principle of comparing within-group variance and between-group variance in ANOVA. The P value is used to indicate whether there is a significant difference between groups. If $P < 0.05$, the dissimilarities within- and between-group were considered to be significantly different [7, 8]. As shown in Fig. 3B, the green box shows the Bray-Curtis distance of the microbial communities among individuals between different groups, while the red and blue boxes reflect the distance among individuals within the same groups. As the $R > 0$ and $P < 0.05$, we said that the distance, namely dissimilarity, among individuals within clust1 is larger than that within clust2. We have added the details for ANOSIM in the revised manuscript. (Line 399-403)

6. Line 142-144, "a prediction model consisted of 6 favorable species, 26 risk species, and age at diagnosis was constructed". The abundance of 193 bacterial species was associated with OS. How did the 6 favorable and 26 risk species were picked?

Response: We appreciate your raising this important point. To investigate whether intratumoral microbes have prognostic value, species significantly associated with OS were served as potential prognostic signatures for further analysis. We divided all patients into training and validation sets according to 8:2. In the training set, Lasso-penalized Cox regression analysis was performed to screen prognostic species using the "glmnet" R package (v4.1-2), which could avoid overfitting and reduce multicollinearity. Tenfold cross-validations were performed to define the optimal value of the lambda penalty parameter; this resulted in the weight of most of the potential prognostic species decreasing to zero, and 6 favorable and 26 risk species with a weight of nonzero remained. Lastly, multivariate Cox regression analysis was used to construct a microbes-based prediction model with the "rms" R package.

Reference

1. Becker, L., et al., *Age-dependent shift in macrophage polarisation causes inflammation-mediated degeneration of enteric nervous system*. Gut, 2018. **67**(5): p. 827-836.
2. Liang, L., et al., *Integration of scRNA-Seq and Bulk RNA-Seq to Analyse the Heterogeneity of Ovarian Cancer Immune Cells and Establish a Molecular*

- Risk Model*. *Frontiers In Oncology*, 2021. **11**: p. 711020.
3. Desbois, M., et al., *Integrated digital pathology and transcriptome analysis identifies molecular mediators of T-cell exclusion in ovarian cancer*. *Nature Communications*, 2020. **11**(1): p. 5583.
 4. Hoadley, K.A., et al., *Cell-of-Origin Patterns Dominate the Molecular Classification of 10,000 Tumors from 33 Types of Cancer*. *Cell*, 2018. **173**(2).
 5. Poore, G.D., et al., *Microbiome analyses of blood and tissues suggest cancer diagnostic approach*. *Nature*, 2020. **579**(7800): p. 567-574.
 6. *Data Types Collected by TCGA*. Available from: <https://www.cancer.gov/about-nci/organization/ccg/research/structural-genomics/tcga/using-tcga/types>.
 7. Wang, J., et al., *Translocation of vaginal microbiota is involved in impairment and protection of uterine health*. *Nature Communications*, 2021. **12**(1): p. 4191.
 8. Anderson, M.J. and D.C.I. Walsh, *PERMANOVA, ANOSIM, and the Mantel test in the face of heterogeneous dispersions: What null hypothesis are you testing?* *Ecological Monographs*, 2013. **83**(4): p. 557-574.

February 22, 2023

Prof. Lei Zhang
Shandong University
Department of Biostatistics
Jinan
China

Re: Spectrum03549-22R1 (The interaction between intratumoral microbiome and immunity is related to the prognosis of ovarian cancer)

Dear Prof. Lei Zhang:

Your manuscript has been accepted, and I am forwarding it to the ASM Journals Department for publication. You will be notified when your proofs are ready to be viewed.

Sincerely,

Se-Ran Jun
Editor, Microbiology Spectrum
